# Assessment of the quality of life of COVID-19 recovered patients at the Ghana Infectious Disease Centre

**Esinam Aku Amedewonu**[1], **Genevieve Cecilia Aryeetey**[2]\*, **Anthony Godi**[3], **Josephine Sackeyfio**[4], **Alfred Dickson Dai-Kosi**[4]

**1** Department of Anaesthesia, Korle Bu Teaching Hospital, Accra, Ghana, **2** Department of Health Policy, Planning and Management, School of Public Health, Colleges of Health Sciences, University of Ghana, Legon, Accra, Ghana, **3** Department of Biostatistics, School of Public Health, University of Ghana, Legon, Accra, Ghana, **4** Department of Community and Preventative Dentistry, University of Ghana Dental School, Korle Bu, Accra, Ghana

\* gcaryeetey@ug.edu.gh

## Abstract

### Background

The Coronavirus Disease (COVID-19), initially thought to be a respiratory disease, is now known to affect multiple organ systems with variable presentation and devastating or fatal complications. Despite the large numbers of people who have suffered this disease globally, the mid- to long-term impact of COVID-19 on a person's general well-being and physical function has not been fully investigated in Ghana.

### Aim

This study sought to determine the Quality of Life (QoL) and associated factors among Ghanaian patients following clinical recovery from COVID-19 infection.

### Methods

This was a cross-sectional quantitative study involving 150 COVID-19 recovered patients attending the review clinic of the Ghana Infectious Disease Centre. Quality of life was estimated using the EuroQol Group Association five-domain, five-level questionnaire (EQ-5D-5L) while participants' overall health status was measured on a visual analogue scale (EQ-VAS): a scale ranging from 0 (worst health) to 100 (best health). Kruskal-Wallis tests were used to assess differences in domain and overall QoL scores while quantile regression was used to determine demographic and clinical factors associated with QoL scores.

### Results

The mean QoL from the EQ-5D-5L assessment tool was (81.5 ± 12.0) %, while the self-reported QoL from the EQ-VAS tool (75.6 ± 22.0) %. Persistence of symptoms after 30 days was significantly associated with EQ-5D-5L QoL (Adjusted median difference [95% CI] = -9.40 [-14.19, -4.61], p<0.001) while access to rehabilitative centres was significantly

**Data Availability Statement:** All relevant data are within the paper and its Supporting Information files. Dataset is available from the Dryad database (DOI: https://doi.org/10.5061/dryad.fttdz091f).

**Funding:** The author(s) received no specific funding for this work.

**Competing interests:** The authors also declare no competing interests in the submission of this manuscript.

associated with EQ-VAS QoL (Adjusted median difference [95% CI] = -29.60 [-48.92, -10.29], p = 0.003).

## Conclusion

Quality of life was relatively good among the COVID-19 recovered patients. Persistence of symptoms and access to rehabilitative centres significantly predicted one's QoL.

## Introduction

The World Health Organization (WHO) on 11th March 2020, declared a global pandemic of COVID-19, in just under three months of the discovery of this disease in the Wuhan Province, China [1,2]. COVID-19 is caused by a virus of the coronaviridae family named Severe Acute Respiratory Syndrome Coronavirus 2 (SARS Cov2) [3,4]. COVID-19 became a major infectious disease which affected people irrespective of their age, sex or location [5,6]. The signs and symptoms are varied but mild to moderate disease often presents with fever, cough, fatigue, dyspnoea, tachypnoea and decreased oxygen saturation, often necessitating hospitalization [6–8]. Again, although the disease was initially thought to be a respiratory disease, COVID-19 has now shown extra-pulmonary manifestations in the renal, gastrointestinal, cardiac, neurological and endocrinological systems [7,9–15]. In mild to moderate disease, patients are expected to recover with discharge home in ten to fourteen days. However, it is now known that some patients may have symptoms persisting for weeks to months following clinical recovery or discharge from isolation—a phenomenon now termed "*Long COVID*" [6,8,16–18]. As of 15th May 2021, it had infected 162,561,793 people globally and resulted in a devastating 3,372,071 mortalities [19,20].

COVID-19 has a multi-dimensional effect on various organs of the body and has also generated a plethora of psychiatric manifestations across the different strata of the society [21]. The importance of disease outcomes in relation to the individual's social integration has been known to interfere with a person's quality of life [21]. Thus, the interest in health-related quality of life of patients who have been affected by the condition increased, during the pandemic and afterwards.

Quality of life (QoL) is the subjective judgement and self-reporting constructed by patients on their individual feelings of well-being to their physical, and social functioning, as well as to their occupational, spiritual, marital and sexual functioning, requiring a degree of cognitive ability [22]. Again, quality of life is a broad concept consisting of medical and psychological aspects which encompasses the standards of health, wealth, comfort, enjoyment, and happiness experienced by an individual or by groups of people including their daily activities, psychological health, physical health status perception, instrumental activities, pain of perception, and their overall satisfaction with their lives [23]. It is a measure of general well-being, the positive and negative features of life, and also a measure of a good life expected by an individual or the society. Values, goals, norms, and the socio-cultural contexts surrounding an individual's life guides these expectations. The World Health Organization (WHO) defines QoL as "an individual's perception of their position in life in the context of culture and value systems in which they live" [23].

Several studies have been conducted to assess the quality of life (QoL) of COVID-19 patients using various QoL assessment tools. For instance, a prospective study carried out by Taboada et al. [24] in 2021 to evaluate the QoL, functional status and persistent symptoms of

patients with COVID-19 induced Acute Respiratory Distress Syndrome (ARDS) at six months post treatment in the ICU. A total of 91 survivors were used. At 6months after ICU discharge, a significant proportion of patients had worsened QoL (67%), persistent symptoms, reduction in the extent of functional status (63%), and persistent functional limitations (45%) compared to their pre-COVID-19 status. Only 16% of the respondents had fully recovered. A limitation for the study was the use of critically ill patients in ICUs of hospitals in only one region.

Another study carried out by Garrigues et al. [25] in 2020 on post discharge persistent symptoms and health-related QoL after hospitalization for COVID-19 revealed that most patients who required hospitalization still had persistent symptoms even after 110 days of discharge especially fatigue (55%) and dyspnoea (42%). Other symptoms noted were however loss of memory (34%), concentration disorder (28%), sleep disorder (30%), and hair loss (20%). Although the HRQoL was quite satisfactory, out of the 56 active workers pre-infection, only 69.1% had resumed work and amongst the 39 patients who had regular sports activity, only 71.8% had resumed physical activity with 46% having a lower level of physical activity.

Lim et al. [17] evaluated the impact of COVID-19 on HRQoL in a multi-ethnic Asian cohort with cardiovascular disease (CVD) in Singapore. Their study reported worsening of HRQoL during the COVID-19 outbreak. There was a significantly associated decline in the psychological health components especially anxiety and depression in patients with pre-existing CVD. A limitation to this study was the fact that it was conducted in Asian patients with CVD hence may not be generalizable to other patient populations or ethnicities.

Further studies conducted by Jacobs et al. [26] on the persistence of symptoms and QoL at 35days post hospitalization for COVID-19 infection reported that 72.7% of their patients had persistent symptoms at day 35. Out of the 183 participants recruited, 52% were employed. At the time the study was conducted, only 29.9% of these cohort had resumed work whiles 73.7% had not on account of persistent symptoms. This study thus revealed that symptoms often persist beyond 35 days of onset of infection, and this usually impairs the individual's ability to perform daily living activities, as well as impairs the QoL, health, mental social, and physical function.

A cross-sectional survey of a geographical cohort carried out by Garrat et al. [27] on non-hospitalized patients concluded that several important dimensions of HRQoL, including general health, social functioning, with role-limitations due to emotional problems and well-being, and aspects of mental health were lower than the general uninfected population norms 1.5–6 months after onset of COVID-19.

While a number of these studies have been carried outside sub-Saharan Africa, few studies exist within the continent since the continent was least affected by the disease in comparison to the more developed countries. Particularly in Ghana, despite a slightly more severe second wave, the mortality has been relatively low with 780 deaths reported as of 30 April 2021, from 92,740 infections and 90,376 recoveries [22]. As a response, the government established the Ghana Infectious Disease Centre (GIDC) as a major referral centre to manage and treat patients. The evidence on how the disease has impacted Ghanaian survivors regarding the mid-to-long-term effects on their health remains limited. This study therefore sought to determine the quality of life of COVID-19 recovered patients attending the post-recovery clinic of the Centre and the factors associated with the quality of life of these patients.

## Materials and methods

### Study setting

The study was conducted at the Ghana Infectious Disease Centre, which is the nation's first infectious disease centre with a 100-bed capacity built in response to the COVID-19 pandemic.

A post-COVID-19 clinic was set up for follow-ups for all COVID-19 recovered patients and averages about 5 patients per day. The first follow-up visit is a week or two after discharge and majority of patients will have 2 to 3 reviews over 6 weeks period. About 80% of the patients who were hospitalized or isolated attended the review clinic. The focus of the clinic is to ascertain and manage the persistence of COVID-19 symptoms, as well as the management of a previously well-controlled co-morbid state which may have worsened due to COVID-19.

## Study population

The study population were all recovered patients (18years and above) who attended the Ghana Infectious Disease Centre (GIDC) post-COVID-19 review clinic and who were able and willing to give informed consent to participate in the study. The inclusion criteria for this study were, Ghanaian adults (18years and above) with laboratory-confirmed diagnoses of COVID-19 who had recovered and were attending the review clinic; Persons who were at least one month in the recovery period and persons who were willing and consented to be part of the study. The exclusion criteria included presence of impairments to reasoning or mental disability such as dementia or Alzheimer's and persons who were not willing and did not consent to be part of the study. Ethical clearance for this study was obtained in August 2021.

## Sample size, sampling and data collection

To determine the sample size, we obtained information on the number of patients who attended the review clinic between January 2021 and June 2021. About 209 patients out of the 798 patients admitted at the Centre were identified. Using Yamane's formula [28], the minimum sample size was estimated as 137 patients. Using an attrition rate of 10% (13 patients), the final sample size obtained was 150 patients.

Purposive sampling was used to recruit the study participants. Clinical and biographic data were extracted from the clinical records of patients. Data extracted included patient's first COVID-19 positive test, negative test, length of stay in hospital or isolation centre, and comorbidities. In addition, the EuroQol Group Association five-domain, five level questionnaire (EQ-5D-5L) [29,30] and the visual analogue scale (EQ-VAS) questionnaire were administered to consenting participants. Data collection, including extraction of data from the clinical records, started on 13[th] August 2021 and ended on 18[th] November 2021. Only the lead author had access to information that could identify individual participants during and/or after data collection.

## Data analysis

The dependent (outcome) variables for the study were QoL scores measured using the EQ-5D-5L and EQ-VAS tools [29,30]. The EQ-5D-5L tool has 5 domains, each with 5 Likert-scale levels indicating quality of life scores in the respective domain (mobility, self-care, usual activities, pain/discomfort, and anxiety/depression) where 1 = best and 5 = worst. A combination of the resulting Likert-scale ratings from the five domains makes up a value set that determines a person's overall quality of life based on comparisons with standardized value sets for different countries.

The values set for Zimbabwe, was used as a proxy for Ghana since it was the only African country with value sets. A single index ranging from <0 to 1.00 was then derived from these value sets, with a value of 1.00 indicating "full health", 0 representing "death" and values <0 indicating "states worse than death" [29,30]. This was converted to percentage scale where 1.00 unit on the original EQ-5D-5L scale = 100% for easy comparison with EQ-VAS which is

also on a percentage scale. The use of the EQ-5D-5L is meant to provide an objective measure of overall quality of life.

Further analysis of the domain scores for comparability with the overall EQ-5D-5L (%) and EQ-VAS (%) scales were achieved by reversing the Likert-scale ratings of the original EQ-5D-5L tool for each domain to 1 = worst to 5 = best. These reversed values were then transformed into the percentage scale using the following formula:

$$Domain\ QoL\ (\%) = \left( \frac{Domain\ Likert\ rating - Lowest\ possible\ rating\ [1]}{Highest\ possible\ rating\ [5] - Lowest\ possible\ rating\ [1]} \right) \times 100$$

Participants also reported their overall health status from their own perspective on a visual analogue scale (EQ-VAS) which is a scale ranging from 0 to 100, where 0 represented their "worst imaginable health" and 100 represented their "best imaginable health status". The use of the EQ-VAS which didn't have any sub-domains is meant to provide an overall measure of quality of life by the study participants which may be subjective. Scores from EQ-VAS (which is a subjective assessment) was used in comparison with scores from EQ-5D-5L (which is an objective measure) to determine if they were correlated.

Frequencies and percentages were used to report categorical outcomes while means and standard deviations were used for continuous outcomes. Kruskal-Wallis tests were used to assess significant differences in domain scores across the categories of variables due to the skewness in the domain specific QoL scores.

Quantile regression models were used to assess the conditional median differences in the two outcomes across the independent variables (sociodemographic, symptoms, comorbidities, complications, treatment modalities and access to rehabilitation post-COVID). This regression method was chosen due to its robustness to some outliers observed in QoL scores which can affect normality and homoscedasticity of residuals from other linear models used to compare means. An alpha level of 5% was set as the significance level for all tests. Data analysis was carried out using Stata version 16.

### Ethical considerations

The research proposal was submitted to the Ghana Health Service Ethics Review Committee to attain ethical approval to undertake the study (Approval ID: GHS–ERC 029/07/21). A permission letter was also provided by the School of Public Health to the Ghana Infectious Disease Centre (GIDC) for study site approval. Informed consent was obtained from each participant through a written consent form attached to each questionnaire.

## Results

### Background characteristics

The mean age of respondents was 43.3 ± 14.8 years with just over a third of participants (35.3%) being 30–39 years (Table 1).

There were more females (54.7%) than males, the dominant religion was Christianity (90%) and most of them (46%) were of Akan ethnicity. Majority of respondents resided in urban areas (94.7%) with at least tertiary level education (83.3%) and formal employment (71.3%). More than half of the respondents (52.7%) had persistence of symptoms beyond thirty days post-recovery with easy fatiguability and headaches being the commonest symptoms (Table 2).

Sixty-three participants (42%) had comorbidities prior to the COVID-19 infection with hypertension being the commonest prior comorbidity (65.1%). Thirty-five participants (23.3%) noted worsening of their comorbidities following the COVID-19 disease with 73.7%

**Table 1. Demographic characteristics of the study participants.**

|  | N | % |
|---|---|---|
| Age (years), Mean = 43.3, SD = 14.8 | | |
| <30 | 21 | 14.0 |
| 30–39 | 53 | 35.3 |
| 40–49 | 30 | 20.0 |
| 50–59 | 23 | 15.3 |
| 60+ | 23 | 15.3 |
| Sex | | |
| Male | 68 | 45.3 |
| Female | 82 | 54.7 |
| Religion | | |
| Christian | 135 | 90.0 |
| Muslim | 8 | 5.3 |
| Other | 7 | 4.7 |
| Ethnicity | | |
| Ga-Dangme | 32 | 21.3 |
| Akan | 69 | 46.0 |
| Ewe | 29 | 19.3 |
| Northern tribe | 12 | 8.0 |
| Other | 8 | 5.3 |
| Place of residence | | |
| Urban | 142 | 94.7 |
| Non-urban | 8 | 5.3 |
| Educational level | | |
| Primary | 5 | 3.3 |
| Secondary | 11 | 7.3 |
| Tertiary | 125 | 83.3 |
| Vocational/Technical | 9 | 6.0 |
| Occupation | | |
| Student | 3 | 2.0 |
| Formally employed | 107 | 71.3 |
| Informally employed | 23 | 15.3 |
| Unemployed | 8 | 5.3 |
| Retired | 9 | 6.0 |
| Total | 150 | 100.0 |

of diabetics and 58.5% of participants with hypertension finding that management of their previous disease had worsened post-COVID-19 (Fig 1).

## Quality of life of study participants

A third of participants (33.3%) suffered from mild to extreme symptoms of anxiety or depression (Table 3). A similar proportion (32.7%) suffered varying degrees of pain or discomfort and only 16% had problems with self-care.

Overall mean EQ-5D-5L Quality of life measured on the percentage scale was slightly higher (81.5 ± 12.0) % than the self-reported EQ-VAS (75.6 ± 22.0) %, with a wider range of scores observed from the EQ-VAS (Fig 2). In this study, the EQ-5D had good reliability with a Cronbach's alpha of 0.79.

## Quality of life scores across the different domains

There were significant differences in some domain specific mean QoL scores across some background and clinical characteristics of the study participants. For mobility, significant differences were observed across age groups where those in their thirties had the highest mean score, and educational level where those with tertiary education also had the highest mean score.

**Table 2. Persistence of symptoms and complications.**

|  | N | % |
|---|---|---|
| **Persistent symptoms (after 30 days)** | | |
| No | 71 | 47.3 |
| Yes | 79 | 52.7 |
| **Symptoms (that persisted after 30 days) *** | | |
| Easy fatigue | 50 | 63.3 |
| Headache | 37 | 46.8 |
| Muscle ache | 33 | 41.8 |
| Extreme fatigue | 29 | 36.7 |
| Cough | 24 | 30.4 |
| Dyspnoea | 18 | 22.8 |
| Chest pain | 17 | 21.5 |
| Ageusia | 17 | 21.5 |
| Anosmia | 15 | 19.0 |
| Sleep disorder | 13 | 16.5 |
| Fever | 6 | 7.6 |
| Attention Disorder | 4 | 5.1 |
| Rigor | 3 | 3.8 |
| Memory loss | 3 | 3.8 |
| Abdominal pain | 2 | 2.5 |
| Nausea & Vomiting | 2 | 2.5 |
| Diarrhoea | 2 | 2.5 |
| **Complications post-COVID-19 infection** | | |
| No | 115 | 76.7 |
| Yes | 35 | 23.3 |
| **Complications*** | | |
| Post-traumatic stress disorder | 6 | 17.1 |
| Pulmonary embolism | 2 | 5.7 |
| Difficulty mobilizing | 4 | 11.4 |
| Cerebro-vascular accident / Stroke | 1 | 2.9 |
| Deep vein thrombosis | 3 | 8.6 |
| Acute kidney injury | 3 | 8.6 |
| **Total** | 150 | 100.0 |

*Multiple responses were allowed.

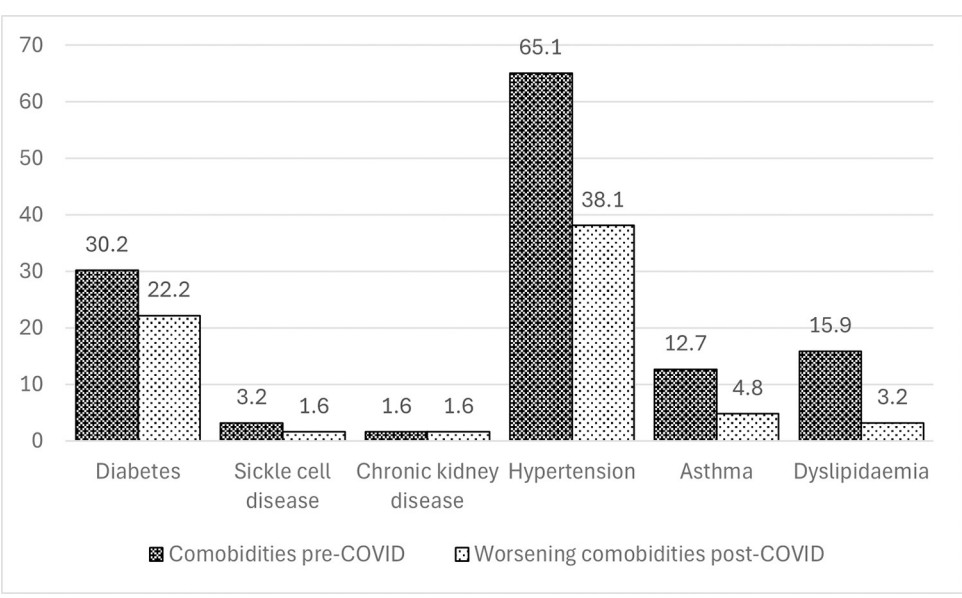

**Fig 1. Comorbidities in study participants.**

**Table 3. Responses to EQ-5D-5L domains.**

|  | N = 150 | % |
|---|---|---|
| Mobility: problems with walking around | 39 | 26.0 |
| Self-care: problems with washing or dishing | 24 | 16.0 |
| Usual activity: problems with usual activity | 40 | 26.7 |
| Pain or discomfort | 49 | 32.7 |
| Anxiety or depression | 50 | 33.3 |

Significant differences were also seen in mobility scores for persistence of symptoms, the presence and worsening of comorbidities, complications resulting from the COVID-19 infection, treatment modalities as well as requirement and access to rehabilitative services.

Participants who did not have their symptoms persistent after 30 days, and those without complications from the COVID-19 infection had significantly higher mean scores regarding their ability to take care of themselves (washing or dressing up). Significant differences are also seen for mean self-care scores across treatment modality and requirement for rehabilitative services post-COVID-19 as seen in Table 4.

On their ability to carry out their usual daily activities, males had a significantly higher mean score, while those with primary education claimed to have absolutely no problems at all, followed by those with tertiary education with the next highest mean score. Significant differences in scores for this domain are seen across persistence of symptoms after 30 days, presence of comorbidities before and after COVID-19 infection, complications and requirement and access to rehabilitative services due to COVID-19.

For the pain/discomfort domain, significant differences were observed for only persistence of symptoms after 30 days, complications post-COVID-19 and access to rehabilitative services where those without such access had a significantly higher mean score. There were significant differences in anxiety/depression domain scores across age groups where those aged 60 years and higher had the best scores with those below 30 years, the lowest. Significant differences

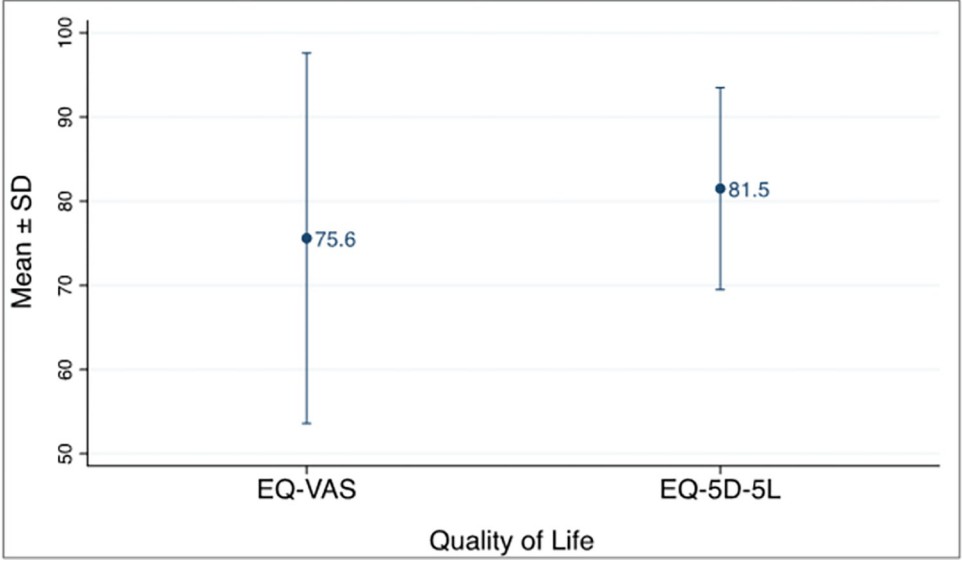

**Fig 2. QoL scores.**

**Table 4. Mean quality of life scores, by EQ-5D-5L domain and background characteristics.**

| | EQ-5D-5L QoL scores, Mean (SD) | | | | | | EQ-VAS QoL scores, Mean (SD) |
|---|---|---|---|---|---|---|---|
| | **Mobility** | **Self-care** | **Usual activities** | **Pain/ Discomfort** | **Anxiety/ Depression** | **Overall** | |
| Age (years) | p<0,001 | p = 0.489 | p = 0.097 | p = 0.498 | p = 0.005 | p = 0.242 | p = 0.017 |
| <30 | 89.3 (18.7) | 97.6 (7.5) | 89.3 (21.8) | 79.8 (30.2) | 78.6 (28.8) | 79.9 (14.8) | 75.2 (21.2) |
| 30–39 | 97.6 (7.4) | 97.2 (9.4) | 92.9 (15.8) | 89.6 (21.1) | 89.6 (17.3) | 84.7 (8.3) | 81.0 (20.5) |
| 40–49 | 93.3 (16.0) | 95.8 (11.5) | 90.0 (23.3) | 85.0 (24.2) | 81.7 (29.3) | 81.7 (13.3) | 76.2 (22.5) |
| 50–59 | 82.6 (28.6) | 89.1 (21.1) | 81.5 (26.3) | 82.6 (26.6) | 88.0 (16.6) | 78.8 (13.5) | 65.2 (27.2) |
| 60+ | 75.0 (23.8) | 89.1 (18.2) | 80.4 (27.1) | 81.5 (25.2) | 96.7 (8.6) | 78.2 (12.1) | 73.3 (16.6) |
| Sex | p = 0.414 | p = 0.183 | p = 0.009 | p = 0.157 | p = 0.319 | p = 0.097 | p = 0.018 |
| Male | 91.9 (16.4) | 96.3 (10.8) | 94.1 (13.7) | 87.5 (24.2) | 89.0 (20.9) | 83.2 (11.4) | 79.9 (19.8) |
| Female | 88.1 (22.0) | 93.0 (15.8) | 83.2 (26.4) | 82.9 (24.8) | 86.0 (22.3) | 80.2 (12.3) | 72.1 (23.1) |
| Educational level | p = 0.041 | p = 0.057 | p = 0.007 | p = 0.251 | p = 0.534 | p = 0.067 | p = 0.017 |
| Primary | 85.0 (33.5) | 90.0 (22.4) | 100.0 (0.0) | 80.0 (32.6) | 90.0 (13.7) | 81.1 (15.4) | 62.0 (34.9) |
| Secondary | 72.7 (36.1) | 81.8 (27.6) | 72.7 (32.5) | 79.5 (31.3) | 90.9 (16.9) | 74.5 (14.2) | 60.0 (29.6) |
| Tertiary | 92.2 (16.0) | 96.2 (10.6) | 90.4 (19.3) | 86.4 (23.7) | 87.4 (22.4) | 82.7 (11.4) | 78.1 (20.1) |
| Vocational/Technical | 80.6 (20.8) | 88.9 (18.2) | 69.4 (34.9) | 75.0 (25.0) | 80.6 (20.8) | 74.0 (12.2) | 67.8 (20.6) |
| Occupation | p = 0.080 | p = 0.157 | p = 0.628 | p = 0.516 | p = 0.205 | p = 0.596 | p = 0.030 |
| Student | 100.0 (0.0) | 100.0 (0.0) | 100.0 (0.0) | 91.7 (14.4) | 91.7 (14.4) | 87.0 (5.2) | 70.0 (17.3) |
| Formally employed | 92.8 (15.0) | 96.5 (10.0) | 89.7 (19.4) | 85.3 (24.8) | 85.5 (22.8) | 82.3 (11.6) | 78.0 (21.2) |
| Informally employed | 79.3 (29.8) | 87.0 (23.7) | 81.5 (29.4) | 78.3 (27.5) | 89.1 (21.1) | 77.1 (13.7) | 62.2 (28.1) |
| Unemployed | 84.4 (29.7) | 90.6 (18.6) | 84.4 (35.2) | 93.8 (11.6) | 90.6 (18.6) | 81.9 (12.5) | 78.8 (5.8) |
| Retired | 83.3 (21.7) | 91.7 (12.5) | 86.1 (22.0) | 88.9 (25.3) | 100.0 (0.0) | 81.5 (12.5) | 81.1 (8.9) |
| Persistent symptoms (after 30 days) | p<0.001 | p<0.001 | p<0.001 | p<0.001 | p<0.001 | p<0.001 | p<0.001 |
| None | 98.6 (5.8) | 99.6 (3.0) | 98.2 (7.7) | 94.7 (15.8) | 95.8 (10.3) | 87.8 (5.1) | 84.5 (13.8) |
| Present | 82.0 (24.0) | 89.9 (17.7) | 79.1 (26.7) | 76.3 (27.7) | 79.7 (26.0) | 75.9 (13.5) | 67.6 (24.8) |
| Comorbidities prior to COVID-19 | p<0.001 | p = 0.313 | p = 0.041 | p = 0.056 | p = 0.387 | p = 0.012 | p<0.001 |
| None | 95.4 (12.4) | 96.0 (10.7) | 91.4 (19.0) | 87.9 (23.1) | 87.4 (24.1) | 83.2 (11.5) | 80.2 (19.6) |
| Present | 82.1 (24.8) | 92.5 (17.2) | 83.7 (25.5) | 81.0 (26.1) | 87.3 (17.9) | 79.3 (12.3) | 69.3 (23.6) |
| Comorbidities post-COVID-19 | p<0.001 | p = 0.376 | p = 0.017 | p = 0.057 | p = 0.875 | p = 0.024 | p<0.001 |
| Did not worsen | 93.9 (14.3) | 95.4 (11.7) | 90.9 (19.4) | 87.0 (23.5) | 87.2 (22.5) | 82.7 (11.5) | 79.7 (17.9) |
| Worsened | 76.4 (27.7) | 91.4 (19.1) | 79.3 (28.1) | 78.6 (27.2) | 87.9 (18.6) | 77.7 (12.9) | 62.1 (28.1) |
| Complications post-COVID-19 | p<0.001 | p<0.001 | p<0.001 | p = 0.002 | p = 0.001 | p<0.001 | p<0.001 |
| None | 94.6 (13.6) | 96.5 (10.9) | 92.4 (18.2) | 88.9 (20.7) | 90.4 (18.9) | 84.2 (9.6) | 79.9 (19.1) |
| Present | 74.3 (27.4) | 87.9 (19.5) | 74.3 (28.1) | 72.1 (31.4) | 77.1 (26.7) | 72.8 (14.8) | 61.4 (24.9) |
| Treatment modality | p<0.001 | p = 0.025 | p = 0.060 | p = 0.557 | p = 0.853 | p = 0.047 | p<0.001 |
| Hospital admission | 77.7 (27.5) | 89.2 (20.0) | 81.8 (27.4) | 82.4 (26.3) | 88.5 (19.2) | 78.3 (12.9) | 66.6 (23.0) |
| Self-isolation at home | 93.8 (14.4) | 96.2 (10.7) | 90.3 (19.9) | 85.8 (24.1) | 86.9 (22.4) | 82.6 (11.5) | 78.6 (20.9) |
| Rehabilitative services post-COVID-19 | p = 0.001 | p = 0.024 | p = 0.006 | p = 0.656 | p = 0.950 | p = 0.048 | p = 0.007 |
| Not required | 92.8 (14.3) | 95.6 (11.8) | 90.3 (19.7) | 85.6 (24.0) | 87.1 (22.2) | 82.3 (11.6) | 77.9 (19.5) |
| Required | 68.1 (35.2) | 86.1 (23.0) | 72.2 (32.0) | 80.6 (29.1) | 88.9 (17.6) | 75.8 (13.6) | 58.9 (31.1) |
| Access to rehabilitative centre | p = 0.001 | p = 0.673 | p = 0.001 | p = 0.017 | p = 0.425 | p = 0.011 | p = 0.008 |
| Did not have access | 91.3 (18.2) | 94.5 (14.0) | 89.7 (20.9) | 86.2 (24.0) | 87.6 (21.7) | 82.0 (12.0) | 76.8 (21.4) |
| Had access | 66.7 (28.0) | 94.4 (11.0) | 63.9 (28.3) | 66.7 (28.0) | 83.3 (21.7) | 74.2 (10.0) | 56.7 (23.0) |
| Total | 89.8 (19.7) | 94.5 (13.9) | 88.2 (22.2) | 85.0 (24.6) | 87.3 (21.6) | 81.5 (12.0) | 75.6 (22.0) |

P-values from Kruskal-Wallis tests used to assess significant differences in domain scores across the categories of variables.

were also seen in this domain for persistence of symptoms after 30 days and complications post-COVID-19 as shown in Table 4.

For the overall EQ-5D-5L QoL, significant differences were seen for persistence of symptoms after 30 days, comorbidities, complications, and rehabilitative services where those without them had higher QoL scores. For EQ-VAS QoL, significant differences were seen across categories of all the variables in Table 4. Those who carried out self-isolation at home as a treatment modality had higher QoL scores for both EQ-5D-5L and EQ-VAS.

## Factors affecting quality of life of recovered patients

The adjusted quantile regression analysis, used to determine the joint factors affecting quality of life showed no differences in median EQ-5D-5L QoL scores for older participants and this was rather decreasing for EQ-VAS though not significantly. Those with secondary, tertiary, and vocational/technical levels of higher education had lower median EQ-5D-5L QoL and EQ-VAS scores compared to those with primary education, but this wasn't significant for both QoL scales.

Across the different categories of occupation, there were no differences in adjusted median EQ-5D-5L QoL scores whereas for EQ-VAS, students had lower median scores compared to the other forms of occupation, though not significantly. Those whose symptoms persisted after 30 days had a significantly lower median QoL score from the EQ-5D-5L tool but not for EQ-VAS after controlling for the other variables in the regression. The presence of comorbidities prior to COVID-19 did not play a significant role in the adjusted models.

Participants who reported worsened comorbidities worsened post-COVID-19 had marginally lower adjusted median QoL score on both the EQ-5D-5L and EQ-VAS tools. There were no significant reductions in median QoL scores for those who reported complications post-COVID-19 from both tools after controlling for the effects of the other variables in the adjusted models. Treatment modality did not play a significant role in differences observed for overall QoL from both tools although for the EQ-VAS scale, those who isolated at home had a 5.63% higher median score compared to those who required hospital admission at some point.

Those who required rehabilitative services post-COVID-19 had a higher median EQ-VAS QoL score, though not significant. Those with access to a rehabilitation centre had a significantly lower median EQ-VAS QoL score while such access did not play a significant role for the EQ-5D-5L QoL scores as seen from Table 5.

## Discussion

The showed that the overall mean EQ-5D-5L (81.5% ± 12.0%) and EQ-VAS (75.6% ± 22.0%) QoL scores for the COVID-19 recovered participants in this study were largely similar to scores obtained in similar studies by Garratt et al and Garrigues et al. [25,27]. It also corresponded to results obtained in a study by Garrigues et al. [25] on post-discharge persistent symptoms and health-related quality of life after hospitalization for COVID-19 which reported an overall mean EQ-5D index value of (86.0 ± 20) % overall mean EQ-VAS score of (70.3 ± 21.5) %. This same study also reported similar mean EQ-5D index value of (82.0 ± 21) % and mean EQ-VAS score of (71.7 ± 22.2) % for patients who were admitted to the ICU.

Furthermore, a multi-ethnic Asian study by Lim et al. [17] on the impact of COVID-19 on health-related quality of life in patients with cardiovascular disease also reported similar findings with mean EQ-VAS and mean EQ-5D index values of (78.6 ± 12.6) % and (89.8 ± 20.0) % respectively. This may possibly be explained by a rapid global response by local governments and international bodies such as WHO, UNICEF, and IMF to provide support, financial relief, and public health assistance to curb the devasting nature of the disease [31,32].

There were significant differences in EQ-VAS scores across age however, when other factors were adjusted for, age ceased to be a significant factor of quality of life for both EQ-5D-5L and EQ-VAS scores. This concurs partially with literature that age is associated with health-related quality of life. This can probably be explained by ageing having a direct relationship with the increasing incidence of comorbidities and problems in patients. For instance, a study by Arab-Zozani et al. [33] reported older patients having lower health-related QoL scores compared with younger patients confirming that COVID-19 impacts older patients more. Other

**Table 5. Adjusted quantile regression of factors influencing EQ-5D-5L and EQ-VAS Quality of Life scores among respondents.**

| | EQ-5D-5L (% scale) | | | | EQ-VAS (% scale) | | | |
|---|---|---|---|---|---|---|---|---|
| | Adjusted | 95% CI for Coeff. | | | Adjusted | 95% CI for Coeff. | | |
| | Coeff. | Lower | Upper | P-value | Coeff. | Lower | Upper | P-value |
| Age (years) | 0.00 | -0.22 | 0.22 | 1.000 | -0.12 | -0.46 | 0.22 | 0.482 |
| Sex | | | | | | | | |
| Male | Ref | -4.93 | 3.53 | 0.744 | Ref | -11.47 | 1.86 | 0.157 |
| Female | -0.70 | | | | -4.80 | | | |
| Educational level | | | | 0.093 | | | | 0.368 |
| Primary | Ref | -20.68 | 8.08 | | Ref | -37.44 | 7.83 | |
| Secondary | -6.30 | -14.61 | 12.01 | | -14.81 | -23.30 | 18.58 | |
| Tertiary | -1.30 | -27.92 | 1.72 | | -2.36 | -29.17 | 17.48 | |
| Vocational/Technical | -13.10 | | | | -5.85 | | | |
| Occupation | | | | 1.000 | | | | 0.532 |
| Student | Ref | -16.06 | 14.66 | | Ref | -8.28 | 40.08 | |
| Formally employed | -0.70 | -17.60 | 15.00 | | 15.90 | -16.42 | 34.89 | |
| Informally employed | -1.30 | -19.35 | 17.95 | | 9.23 | -17.01 | 41.69 | |
| Unemployed | -0.70 | -20.25 | 18.85 | | 12.34 | -11.25 | 50.29 | |
| Retired | -0.70 | | | | 19.52 | | | |
| Had persistent symptoms (after 30 days) | -9.40 | -14.19 | -4.61 | <0.001 | -4.54 | -12.07 | 3.00 | 0.236 |
| Had comorbidities prior to COVID-19 | -0.00 | -5.66 | 5.66 | 1.000 | -2.12 | -11.03 | 6.79 | 0.639 |
| Comorbidities worsened post-COVID | -0.00 | -7.08 | 7.08 | 1.000 | -2.07 | -13.21 | 9.07 | 0.714 |
| Had complications post-COVID | -1.50 | -7.59 | 4.59 | 0.627 | -4.54 | -14.12 | 5.04 | 0.350 |
| Treatment modality | | | | | | | | |
| Hospital admission | Ref | -6.79 | 5.39 | 0.821 | Ref | -3.95 | 15.22 | 0.247 |
| Self-isolation at home | -0.70 | | | | 5.63 | | | |
| Rehabilitative services required post-COVID | 0.00 | -9.04 | 9.04 | 1.000 | 12.5 | -1.72 | 26.73 | 0.084 |
| Had access to rehabilitative centre | -0.70 | -12.97 | 11.57 | 0.910 | -29.60 | -48.92 | -10.29 | 0.003 |

Adjusted Coeff: Adjusted or conditional median differences. CI: Confidence Interval.

studies also reported similar findings in patients older than 60 years having severer disease thereby necessitating ICU admission confirming poorer quality of life as age increases [34].

Also, women in this study had lower mean QoL scores for both scales but this was not a significant factor when other factors were controlled for in the adjusted regression analysis. Studies by Jacobs et al. and Arab-Zozani et al. [26,33] have revealed sex to be a significant factor. Literature also suggests women being more susceptible to experiencing more somatic symptoms due to physiological and socialization factors [35]. In contrast to this study, other studies have reported an equal prevalence of COVID-19 in both sexes with males experiencing a greater severity of disease [36,37] probably due to behaviour, intrinsic risk, and exposure [6].

There were significant differences in mean QoL scores across educational level as well, with tertiary education having the highest mean scores but this relationship was however not significant when other factors were considered in the adjusted analysis. This may be as a result of a better understanding of COVID-19 and measures to manage the disease when education is considered independently of the other factors [33,38]. This differed from some reports which showed higher levels of education being associated with lower mean health related QoL scores as a result of higher education being associated with higher levels of concern about the novel virus as well as greater levels of awareness about the COVID-19 pandemic and its impact on life. [33] Another study by Nguyen et al. [38] also observed high prevalence of depression among patients with university/college education as a result of the stress from the COVID-19 pandemic which affected their health-related quality of life.

This study had majority of respondents being formally employed. Occupation by itself did not have any association with EQ-5D-5L QoL and for EQ-VAS QoL, this relationship was also not significant when other factors were considered from the regression analysis although there were significant differences in mean EQ-VAS QoL scores across occupation, with those formally employed having higher QoL scores compared to the informally employed, unemployed or retired. It corresponded to findings from literature reporting higher scores for employed patients. This could be due to employed patients having the opportunity to obtain better healthcare from their regular source of income which they had not lost during the pandemic. [39] One reason suggested is that employed patients can obtain better healthcare from their regular source of income which they did not lose during the pandemic [39]. On the other hand, the informally employed, who often live 'hand-to-mouth' in Sub-Saharan Africa, may have faced greater hardship from the brief lockdowns and restrictions in movement caused by the pandemic.

The presence of persistent symptoms was significantly associated with reduced EQ-5D-5L QoL % scores from the regression analysis and this corresponded with findings from literature reporting the presence of symptoms such as fatigue, muscle weakness, anxiety, depression and sleep difficulties as long as 6 months after onset of symptoms thereby affecting health related quality of life. [40] Garrigues et al. [25] reported on persistent symptoms (fatigue, dyspnoea, memory loss, concentration and sleep disorders) 110 days after hospital discharge affecting patients quality of life in their study on post-discharge persistent symptoms and health-related quality of life after hospitalization for COVID-19. Garratt et al. [27] also reported in their study on the quality of life after COVID-19 without hospitalization with prolonged fatigue being a symptom for non-hospitalised patients several months after their diagnosis. A study by Jacobs et al. [26] on the persistence of symptoms and quality of life 35 days after hospitalization for COVID-19 infection also reported fatigue and dyspnoea as the most persistent symptom at day 35, greatly impacting the quality of life of patients with difficulty walking, lifting, and walking up stairs.

Majority of respondents had hypertension (65.1%) or diabetes (30.2%) before contracting COVID-19. Having comorbidities prior to COVID-19 infection was found to be statistically significant with participants reporting lower mean quality of life (EQ-5D-5L and EQ-VAS) % scores but not from the regression analysis adjusted for other factors. Studies conducted by Lim et al. [17] in Asians with pre-existing cardiovascular disease during the COVID-19 pandemic reported significant decline in the health-related quality of life (HRQoL) and psychological components of patients. Arab-Zozani et al. [33] also reported lower HRQoL scores in patients with diabetes. This may probably be because COVID exacerbates symptoms especially diabetes symptoms resulting in severer disease [15]. Additionally, patients with comorbidities constitute the vulnerable groups with lower mean quality of life scores during the pandemic [38].

A study by Zhou et al. [15] on diabetic patients with COVID-19 reported abnormal pre- and post-prandial blood glucose levels in these patients leading to higher risk of secondary infection and mortality. Poor glycaemic control has been associated with poorer outcomes in diabetic patients [41]. These findings are also consistent with research conducted in the US on COVID-19 patients in critical care reporting patients with comorbidities such as hypertension, diabetes and cardiovascular disease being admitted more into intensive care units (ICU) as a result of severer disease leading to lower quality of life scores [34].

About a quarter (23%) of respondents suffered complications due to COVID-19 infection. Such complications included post-traumatic stress disorder, difficulty mobilizing, deep vein thrombosis, acute kidney injury, pulmonary embolism, newly diagnosed diabetes and hypertension, migraine, and stroke. These were significantly associated with lower mean quality of

life scores but were not significant from the adjusted regression analysis. Some researchers have investigated the long-term sequelae of extrapulmonary manifestations during follow-up. They reported complications such as persistent renal dysfunction, newly diagnosed diabetes, venous thrombo-embolism, cardiovascular events, and cerebrovascular events [40] and also revealed the mental health of those participants as poor or fair. The development of complications post-COVID-19 infection affects the mental and psychological health and wellbeing of patients consequently affecting their health-related quality of life [42].

In this study, a third of respondents reported having anxiety or depression (33.33%) and symptoms of pain and discomfort (32.67%). COVID-19 is a novel disease and arrived with a lot of confusion and uncertainty. Initial reports showed high mortality and government responses were initially erratic with restriction of movement, travel, and lockdowns [1,43–45]. Public socializing and close human interactions were restricted and offices, businesses and even shopping centres and marketplaces had to be closed. The mental and psychological impact on individuals was great and this is clearly reflected in the high levels of anxiety, depression, discomfort, and pain in this study. Several respondents suffered from mild to extreme symptoms of anxiety or depression or had mild to extreme levels of pain and discomfort. However, majority (88.0%) were not offered rehabilitative services (psychotherapy, chest, and limb physiotherapy) during or after recovery from COVID-19.

Comparison of mean EQ-5D-5L and EQ-VAS scores showed significantly higher scores for those offered rehabilitation but when other factors were considered in the adjusted regression analysis, these relationships were no longer significant. Rehabilitative services were offered only to few hospitalized patients who had severe to critical manifestations of COVID-19. Majority of the participants who had mild to moderate disease were not offered rehabilitative services. Some patients who were not offered such services commented on their desire for it. Some also reported seeking personal physiotherapists and psychologist services at their homes to help them through the recovery process.

Ninety four percent (94%) of the respondents had no access to rehabilitative centres. A few reasons for this being that they were not told, such centres were not available where they lived, some complained it was too expensive, too far away, or didn't know where to assess such services. Out of those that had access to rehabilitative centres over three-quarters (77.8%) benefited from psychotherapy (psychologist review).

A study conducted by Lemhöfer et al. [46] assessing the rehabilitation needs of patients during and after COVID-19 reported the disease causing a wide range of problems affecting several organ systems and persisting for long periods of time, requiring long-term rehabilitation needs. Many other studies [38,47] have also researched into the health-related quality of life of COVID-19 patients and reported on the need for multi-disciplinary rehabilitative therapies to treat the wide range of symptoms and functional deficits, especially in the acute phase of the disease [48,49].

Rehabilitation is an extremely important and essential component of the treatment and recovery process of COVID-19 patients and needs to be initiated during all phases of the disease process to enable patients return to normal functionality early [50] .There is a need for a multi-disciplinary rehabilitative therapy to treat the wide range of symptoms and functional deficits, especially in the acute phase of the disease [38,47–49]. COVID-19 disease has been reported to cause a wide range of problems affecting several organ systems and persisting for long periods of time, requiring long-term rehabilitation needs [46]. Provision of these services remains a challenge in countries like Ghana where availability of and access to the services, number of qualified personnel, and cost of services are inhibitory factors. Those who received rehabilitation services at the centre and were advised to continue elsewhere upon discharge,

did not have access to rehabilitation centres due to various reasons such as distance, expense, and lack of accessibility.

Some limitations of the study include first, the possibility of recall bias by participants and the use of a single treatment centre which may not be indicative of the national circumstances. Second, the sample may have appeared skewed to those with higher level of education. Plausible reason could be that most of the cases that showed up the Centre, which is a tertiary level facility, were referrals from other facilities typically used by populations with higher socio-economic status and they are more likely to have higher levels of education. Third, the novel nature of COVID-19 with peculiar outcomes for different segments of the population coupled with lack of a large set of patient's post-hospitalization within the general population meant sampling options were limited and so results may not be generalizable to the entire population. However, the study provides a strong basis for further investigations into the influence of COVID-19 on survivors. Finally, since COVID-19 is a novel disease, the knowledge on its pathology, management and complications is currently evolving. It is possible that findings of this study may change (such as the level of quality of life) as there is better understanding of the disease.

## Conclusion

Overall, this study showed that majority of respondents maintained a relatively good quality of life after surviving COVID-19. For the objective assessment of QoL using the EQ-5D-5L tool, persistence of symptoms after 30 days, comorbidities and complications prior and post-COVID-19, treatment modality and requirement and access to rehabilitative services were significant independent factors but after controlling for these variables in adjusted regression analysis, persistence of symptoms remained significant and negatively affecting this QoL.

For the self-reported EQ-VAS assessment of QoL, age, sex, education, occupation, persistence of symptoms after 30 days, comorbidities and complications prior and post-COVID-19, treatment modality and requirement and access to rehabilitative services were significant independent factors but after controlling for them in adjusted regression analysis, access to rehabilitative services surprisingly remained significant and negatively affecting this subjective measure of QoL.

## Supporting information

**S1 Dataset.**
(XLSX)

## Acknowledgments

1. Levi Nii Ayi Ankrah, *National Reconstructive Plastic Surgery & Burns Centre*, *Korle Bu Teaching Hospital*, *Accra*, *Ghana*

2. Thomas Ndanu, *Department of Community and Preventative Dentistry*, *University of Ghana Dental School*, *Korle Bu*, *Accra*, *Ghana*

3. Dr. Oliver-Commey and all the staff of Ghana Infectious Disease Centre (GIDC)

4. All the patients of the post-Covid-19 review clinic.

## Author Contributions

**Conceptualization:** Esinam Aku Amedewonu, Josephine Sackeyfio.

**Data curation:** Esinam Aku Amedewonu.

**Formal analysis:** Esinam Aku Amedewonu, Anthony Godi.

**Funding acquisition:** Esinam Aku Amedewonu.

**Investigation:** Esinam Aku Amedewonu.

**Methodology:** Esinam Aku Amedewonu.

**Project administration:** Esinam Aku Amedewonu.

**Resources:** Esinam Aku Amedewonu.

**Software:** Esinam Aku Amedewonu, Anthony Godi.

**Supervision:** Esinam Aku Amedewonu, Genevieve Cecilia Aryeetey.

**Validation:** Esinam Aku Amedewonu, Genevieve Cecilia Aryeetey, Anthony Godi.

**Visualization:** Esinam Aku Amedewonu, Genevieve Cecilia Aryeetey, Anthony Godi, Josephine Sackeyfio.

**Writing – original draft:** Esinam Aku Amedewonu, Anthony Godi, Josephine Sackeyfio.

**Writing – review & editing:** Esinam Aku Amedewonu, Genevieve Cecilia Aryeetey, Anthony Godi, Josephine Sackeyfio, Alfred Dickson Dai-Kosi.

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
