## [Decision Letter · Decision Letter 0]

26 Dec 2023

PONE-D-23-33323Assessment of the Quality of Life of COVID-19 recovered patients at the Ghana Infectious Disease CentrePLOS ONE

Dear Dr. Aryeetey,

Thank you for submitting your manuscript to PLOS ONE. After careful consideration, we feel that it has merit but does not fully meet PLOS ONE’s publication criteria as it currently stands. Therefore, we invite you to submit a revised version of the manuscript that addresses the points raised during the review process.

We look forward to receiving your revised manuscript.

Kind regards,

Daniel Ahorsu, PhD

Academic Editor

PLOS ONE

Journal Requirements:

"None"

Reviewers' comments:

Reviewer's Responses to Questions

**Comments to the Author**

1. Is the manuscript technically sound, and do the data support the conclusions?

Reviewer #1: No

Reviewer #2: Yes

Reviewer #3: Partly

2. Has the statistical analysis been performed appropriately and rigorously? 

Reviewer #1: No

Reviewer #2: Yes

Reviewer #3: Yes

3. Have the authors made all data underlying the findings in their manuscript fully available?

Reviewer #1: No

Reviewer #2: Yes

Reviewer #3: Yes

4. Is the manuscript presented in an intelligible fashion and written in standard English?

Reviewer #1: No

Reviewer #2: Yes

Reviewer #3: Yes

5. Review Comments to the Author

Reviewer #1: Thank you for the opportunity to review this article titled "Assessment of the Quality of life of COVID-19 recovered patients at the Ghana Infectious Diseases Centre". The researchers had a well-defined aim to investigate the impact of COVID-19 on Quality of life of recovered patients. The selected tool for assessing quality of life was also appropriate. Finally the use of linear regression adds dept to the study.

However, I am unable to recommend this paper for publication due to the following reasons:

1. The introduction lacks dept and organization. Researchers should include relevant related literature on similar study done in other parts of the world and also introduce subheadings to improve readability.

2. The writing style will benefit from improvement in areas of conciseness and flow of ideas.

3. The researchers did not state any inclusion or exclusion criteria for the sample selected. Even though patients were all admitted for COVID-19 facility, their length of stay and COVID-19 related complications were not the same .

4. The researchers did not explain if any power analysis was done to arrive at the sample size.

5. Also the sample selected is skewed towards formally employed (71%) and individuals with tertiary level of education (83%) and thus not representative of the population.

6. The researchers mentioned recall bias but did not state explicitly the data collection process and timelines.

7. Finally they failed to mention limitations associated with the research design used and its impact on research results and conclusions

Reviewer #2: 1.Title is reflective of study- objectives and rationale

2.Sample size of 150 adequate for a clinical sample

3.No reported Cronbach’s alpha for the (EQ-VA) visual analogue scale

4.No in text references for Discussion Lines 6,7 and 8 - global response by local governments and international bodies such as WHO, UNICEF, and IMF to provide support, financial relief, and public health assistance to curb the devastating nature of the disease.

5.Discussion Lines 29-30 Women may also be more worried when it comes to disease with a decreased ability of coping- This needs evidence to back it.

Reference list: Rearrange 2, 11, 12, 15, 18, 20, 21, 22, 32, 33, 35. 36

Reviewer #3: This study addresses an important but often overlooked area in health research. However, there are some issues that need to be addressed before this paper can be published.

Abstract

• “Differences were not significantly observed for factors such as

marital status, residence, ethnicity, and access to rehabilitative centres” does not read well. I suggest “Significant differences were not observed in quality of life for factors such as marital status, residence, ethnicity, and access to rehabilitative centres”

Introduction

• The introduction is fine with some edits. For example, the sentence in lines 79-82 is unclear in terms of comparison with NCDs and genetic diseases.

• The background could touch on some of the potential predictors rooted in relevant literature and possibly a theory relevant to QOL and its predictors.

• The specific objectives addressed in this study need to be stated.

Materials and Methods

Lines 154-156. “Linear regression models were used to assess the mean differences” Linear regression models are used to test relationships/predictions. Check and clarify.

Results and Discussion

• Authors need to justify the analysis in Table 5 as it is a repetition of analysis presented in Table 4.

• The discussion needs to be rewritten to focus on the key findings that addressed the objectives of the study.

• No contextual reasons were adduced for the high QOL scores among the study participants beyond global and local efforts (which were not duly discussed). What about possible roles of family and significant others in the Ghanaian context of care?

• “This study found four main predictors of quality of life in the Ghanaian patient post-recovery from COVID-19. These were sex, persistent symptoms, worsening comorbidities, and development of complications post-COVID-19 infection”. This is not accurate as there are differences in QOL across several factors.

• The discussion needs to be focused and structured according to the study objectives.

6. PLOS authors have the option to publish the peer review history of their article (what does this mean?). If published, this will include your full peer review and any attached files.

Reviewer #1: No

Reviewer #2: **Yes: **Eric Howusu-Kumi

Reviewer #3: No

---

## [Author Response · Author response to Decision Letter 0]

29 Feb 2024

The Editor

PLOS ONE

Dear Editor,

Response to Reviewers: Assessment of the Quality of Life of COVID-19 recovered patients at the Ghana Infectious Disease Centre

We are grateful to the editor and reviewers for their feedback and helpful recommendations. We are glad to have the chance to resubmit an improved version of our paper for your further evaluation. In the following, we address the reviewers’ remarks and indicate the relevant modifications we applied to the paper. The line numbers we refer to are from the revised manuscript with the track changes off.

Reviewer #1:

1. The introduction lacks dept and organization. Researchers should include relevant related literature on similar study done in other parts of the world and also introduce subheadings to improve readability.

Response: The introduction has been revised to reflect the depth and inclusion of relevant studies. We however used paragraphs to introduce issues instead of the sub-headings.

2. The writing style will benefit from improvement in areas of conciseness and flow of ideas.

Response: This has carefully been looked at and improved.

3. The researchers did not state any inclusion or exclusion criteria for the sample selected. Even though patients were all admitted for COVID-19 facility, their length of stay and COVID-19 related complications were not the same.

Response: The inclusion criteria and exclusion criteria employed has been indicated in lines 143-151 of the revised manuscript (without track changes).

4. The researchers did not explain if any power analysis was done to arrive at the sample size.

Response: We have provided an explanation on the sample size determination in lines 153-157.

5. Also, the sample selected is skewed towards formally employed (71%) and individuals with tertiary level of education (83%) and thus not representative of the population.

Response: Most of the cases that showed up the GIDC which is a (tertiary level facility) were referrals from other facilities typically used by populations with higher socio-economic status and they are more likely to have higher levels of education.

6. The researchers mentioned recall bias but did not state explicitly the data collection process and timelines.

7. 

Response: The data collection process, particularly for the administration of the EuroQoL tools was to recovered patients who have been attending the post-recovery clinic from one month and above. This has been indicated in the inclusion criteria. 

8. Finally, they failed to mention limitations associated with the research design used and its impact on research results and conclusions.

Response: The study’s limitations have been included in lines 453-463 of the revised manuscript.

Reviewer #2:

1. Title is reflective of study- objectives and rationale.

2. Sample size of 150 adequate for a clinical sample.

3. No reported Cronbach’s alpha for the (EQ-VAS) visual analogue scale.

Response: The EQ-VAS is a single measure of quality of life from a self-reported scale and so the use of the Cronbach’s alpha was not applicable unlike the EQ-5D-5L which was derived from multi-dimensional Likert scale measures.

4. No in text references for Discussion Lines 6,7 and 8 - global response by local governments and international bodies such as WHO, UNICEF, and IMF to provide support, financial relief, and public health assistance to curb the devastating nature of the disease.

Response: This has been resolved. (Ref to line numbers 326 – 328)

5. Discussion Lines 29-30 Women may also be more worried when it comes to disease with a decreased ability of coping- This needs evidence to back it.

Response: This statement has been deleted from the manuscript. 

6. Reference list: Rearrange 2, 11, 12, 15, 18, 20, 21, 22, 32, 33, 35. 36

Response: The list of references has been revised.

Reviewer #3:

Abstract

1. “Differences were not significantly observed for factors such as marital status, residence, ethnicity, and access to rehabilitative centres” does not read well. I suggest “Significant differences were not observed in quality of life for factors such as marital status, residence, ethnicity, and access to rehabilitative centres”.

Response: The statement being referred to has been taken out of the abstract

Introduction

2. The introduction is fine with some edits. For example, the sentence in lines 79-82 is unclear in terms of comparison with NCDs and genetic diseases.

Response: The Introduction has been extensively revised.

3. The background could touch on some of the potential predictors rooted in relevant literature and possibly a theory relevant to QOL and its predictors.

Response: The Background of the Introduction has been revised with additional literature to that effect.

4. The specific objectives addressed in this study need to be stated.

Response: This has been included in the introduction (Ref to line numbers 125 – 127)

Materials and Methods

5. Lines 154-156. “Linear regression models were used to assess the mean differences” Linear regression models are used to test relationships/predictions. Check and clarify.

Response: This statement has been deleted from the manuscript. The regression methods used have also changed and explained accordingly.

Results and Discussion

6. Authors need to justify the analysis in Table 5 as it is a repetition of analysis presented in Table 4.

Response: The analysis in Table 4 shows the mean QoL scores and compares them by subdomains of EQ-5D-5L as well as overall with Kruskal-Wallis tests for each independent factor. Table 5 in its current form is from adjusted quantile regression using the overall median QoL scores and the reason for this has been explained in line numbers 265-266.

7. The discussion needs to be rewritten to focus on the key findings that addressed the objectives of the study.

Response: The discussions have been revised accordingly.

8. No contextual reasons were adduced for the high QOL scores among the study participants beyond global and local efforts (which were not duly discussed). What about possible roles of family and significant others in the Ghanaian context of care?

Response: We believe the revised discussions will resolve this.

9. “This study found four main predictors of quality of life in the Ghanaian patient post-recovery from COVID-19. These were sex, persistent symptoms, worsening comorbidities, and development of complications post-COVID-19 infection”. This is not accurate as there are differences in QOL across several factors.

Response: There were multiple factors independently associated with QoL from Table 4, but the analysis shown in Table 5 which takes care of confounding effects using the medians provides a more robust determination of these significant factors.

10. The discussion needs to be focused and structured according to the study objectives.

Response: The discussions have been revised accordingly.

Thank you

---

## [Decision Letter · Decision Letter 1]

3 May 2024

PONE-D-23-33323R1Assessment of the Quality of Life of COVID-19 recovered patients at the Ghana Infectious Disease CentrePLOS ONE

Dear Dr. Aryeetey,

Thank you for submitting your manuscript to PLOS ONE. After careful consideration, we feel that it has merit but does not fully meet PLOS ONE’s publication criteria as it currently stands. Therefore, we invite you to submit a revised version of the manuscript that addresses the points raised during the review process.

We look forward to receiving your revised manuscript.

Kind regards,

Daniel Ahorsu, PhD

Academic Editor

PLOS ONE

Journal Requirements:

Additional Editor Comments:

The authors have significantly revised the manuscript. However, the reviewers have indicated further minor comments. I believe that responding to the comments will further enhance the quality of the manuscript.

In addition to the other comments, the authors should further address this comment by adding a sentence in the limitation section. "5. Also, the sample selected is skewed towards formally employed (71%) and individuals with tertiary level of education (83%) and thus not representative of the population.

Response: Most of the cases that showed up the GIDC which is a (tertiary level facility) were referrals from other facilities typically used by populations with higher socio-economic status and they are more likely to have higher levels of education."

Reviewers' comments:

Reviewer's Responses to Questions

**Comments to the Author**

1. If the authors have adequately addressed your comments raised in a previous round of review and you feel that this manuscript is now acceptable for publication, you may indicate that here to bypass the “Comments to the Author” section, enter your conflict of interest statement in the “Confidential to Editor” section, and submit your "Accept" recommendation.

Reviewer #2: All comments have been addressed

Reviewer #3: All comments have been addressed

2. Is the manuscript technically sound, and do the data support the conclusions?

Reviewer #2: Yes

Reviewer #3: Yes

3. Has the statistical analysis been performed appropriately and rigorously? 

Reviewer #2: I Don't Know

Reviewer #3: Yes

4. Have the authors made all data underlying the findings in their manuscript fully available?

Reviewer #2: Yes

Reviewer #3: Yes

5. Is the manuscript presented in an intelligible fashion and written in standard English?

Reviewer #2: Yes

Reviewer #3: Yes

6. Review Comments to the Author

Reviewer #2: In my opinion, I think the revision I requested for were met. Specifically, issues on Cronbach’s alpha for the (EQ-VAS) visual analogue scale was resolved. The in text references for Discussion Lines 6,7 and 8 has also been resolved. Discussion Lines 29-30 has been deleted. The authors also rearranged the Reference list: 2, 11, 12, 15, 18, 20, 21, 22, 32, 33, 35 and 36

Reviewer #3: Lines 32-33: The sentence does not read well.

Line 42: the beta value should be checked (9.40 [-14.19, -4.61]) because the 95%Cis are all in the negative.

7. PLOS authors have the option to publish the peer review history of their article (what does this mean?). If published, this will include your full peer review and any attached files.

Reviewer #2: **Yes: **Eric Howusu-Kumi

Reviewer #3: No

---

## [Author Response · Author response to Decision Letter 1]

15 May 2024

The Editor

PLOS ONE

Dear Editor,

Response to Reviewers: Assessment of the Quality of Life of COVID-19 recovered patients at the Ghana Infectious Disease Centre

We are grateful to the editor and reviewers for their feedback and helpful recommendations. We are glad to have the chance to resubmit an improved version of our paper for your further evaluation. In the following, we address the reviewers’ remarks and indicate the relevant modifications we applied to the paper. 

Additional editor comments

In addition to the other comments, the authors should further address this comment by adding a sentence in the limitation section. "5. Also, the sample selected is skewed towards formally employed (71%) and individuals with tertiary level of education (83%) and thus not representative of the population.

Response: Most of the cases that showed up the GIDC which is a (tertiary level facility) were referrals from other facilities typically used by populations with higher socio-economic status and they are more likely to have higher levels of education."

Response: We thank the editor for the suggestion. The limitations section has been updated accordingly. Lines 448-452

Reviewer 3

Reviewer #3: Lines 32-33: The sentence does not read well.

Line 42: the beta value should be checked (9.40 [-14.19, -4.61]) because the 95%CIs are all in the negative.

Response: We thank the reviewer for noticing these errors. Lines 32-33 and 42 have been revised. The beta value is -9.40. 

Authors additional revisions

We also included additional information on the sampling section to improve clarity. Lines 148-152

Thank you

---

## [Decision Letter · Decision Letter 2]

11 Jun 2024

Assessment of the Quality of Life of COVID-19 recovered patients at the Ghana Infectious Disease Centre

PONE-D-23-33323R2

Dear Dr. Aryeetey,

We’re pleased to inform you that your manuscript has been judged scientifically suitable for publication and will be formally accepted for publication once it meets all outstanding technical requirements.

Kind regards,

Daniel Ahorsu, PhD

Academic Editor

PLOS ONE

Additional Editor Comments (optional):

Reviewers' comments:

Reviewer's Responses to Questions

**Comments to the Author**

1. If the authors have adequately addressed your comments raised in a previous round of review and you feel that this manuscript is now acceptable for publication, you may indicate that here to bypass the “Comments to the Author” section, enter your conflict of interest statement in the “Confidential to Editor” section, and submit your "Accept" recommendation.

Reviewer #3: All comments have been addressed

2. Is the manuscript technically sound, and do the data support the conclusions?

Reviewer #3: Yes

3. Has the statistical analysis been performed appropriately and rigorously? 

Reviewer #3: Yes

4. Have the authors made all data underlying the findings in their manuscript fully available?

Reviewer #3: Yes

5. Is the manuscript presented in an intelligible fashion and written in standard English?

Reviewer #3: Yes

6. Review Comments to the Author

Reviewer #3: (No Response)

7. PLOS authors have the option to publish the peer review history of their article (what does this mean?). If published, this will include your full peer review and any attached files.

Reviewer #3: No

---

## [Editor Report · Acceptance letter]

9 Jul 2024

PONE-D-23-33323R2 

PLOS ONE

Dear Dr. Aryeetey, 

I'm pleased to inform you that your manuscript has been deemed suitable for publication in PLOS ONE. Congratulations! Your manuscript is now being handed over to our production team.

Kind regards, 

on behalf of

Dr. Daniel Ahorsu 

Academic Editor

PLOS ONE